https://doi.org/10.1038/s41467-021-21104-8　**OPEN**

# Unusual solute segregation phenomenon in coherent twin boundaries

Cong He [1,6], Zhiqiao Li[1,6], Houwen Chen [1,2,3✉], Nick Wilson[4] & Jian-Feng Nie [5✉]

Interface segregation of solute atoms has a profound effect on properties of engineering alloys. The occurrence of solute segregation in coherent twin boundaries (CTBs) in Mg alloys is commonly considered to be induced by atomic size effect where solute atoms larger than Mg take extension sites and those smaller ones take compression sites in CTBs. Here we report an unusual solute segregation phenomenon in a group of Mg alloys—solute atoms larger than Mg unexpectedly segregate to compression sites of $\{10\bar{1}1\}$ fully coherent twin boundary and do not segregate to the extension or compression site of $\{10\bar{1}2\}$ fully coherent twin boundary. We propose that such segregation is dominated by chemical bonding (coordination and solute electronic configuration) rather than elastic strain minimization. We further demonstrate that the chemical bonding factor can also predict the solute segregation phenomena reported previously. Our findings advance the atomic-level understanding of the role of electronic structure in solute segregation in fully coherent twin boundaries, and more broadly grain boundaries, in Mg alloys. They are likely to provide insights into interface boundaries in other metals and alloys of different structures.

---

[1] International Joint Laboratory for Light Alloys (Ministry of Education), College of Materials Science and Engineering, Chongqing University, Chongqing, P.R. China. [2] Electron Microscopy Center, Chongqing University, Chongqing, P.R. China. [3] Shenyang National Laboratory for Materials Science, Chongqing University, Chongqing, P.R. China. [4] CSIRO Mineral Resources, Clayton Vic, Australia. [5] Department of Materials Science and Engineering, Monash University, Vic, Australia. [6] These authors contributed equally: Cong He, Zhiqiao Li ✉email: hwchen@cqu.edu.cn; jianfeng.nie@monash.edu

Interfaces such as twin boundaries and grain boundaries (GBs) are ubiquitous crystal defects in materials[1–6]. Enrichment of solute atoms in such interfaces has been reported to affect many properties of engineering alloys, including GB stability, mechanical properties, and crystallographic texture[2,7–12]. In Mg alloys, representative examples include periodical segregation of solute atoms to coherent twin boundaries (CTBs)[13,14] and solute segregation in general GBs[8,9,15,16]. Gaining a deeper fundamental understanding of the origin of the interface segregation phenomena is important to the rational design of alloys with unprecedented properties.

Twin boundaries are special high-angle GBs, with lower interfacial energies than general GBs. In Mg and its alloys with a close-packed hexagonal (HCP) structure, twin boundaries are generated by twinning during plastic deformation. There are three main types of CTBs that are commonly observed, including $\{10\bar{1}1\}$, $\{10\bar{1}2\}$, and $\{10\bar{1}3\}$[6,13,17–20]. It has been reported[13] that elastic strain exists in these CTBs where compression and extension sites are alternately distributed. The currently accepted notion is that solute atoms larger than Mg segregate favorably to the extension sites, while smaller solutes segregate to the compression sites[13,21–24]. This can be understood by considering a large solute in an Mg lattice, where the solute will displace the surrounding Mg atoms away from their normal lattice positions, creating a local strain. If the solute relocates to an extension site in a CTB, then it is still bonded with Mg, however, strain in the lattice is removed, and the strain in the CTB extension site is reduced. To date, this notion of the elastic strain minimization has successfully explained all the experimentally observed solute segregation behaviors in CTBs in Mg alloyed with a rare-earth (RE), Zn, or Ag, and Mg alloyed with multiple elements, such as RE and Zn, or RE and Ag[13,14,25–27]. Although chemical bonding is reported as another factor that affects solute segregation in ceramic and semiconductor materials[28,29], its effects on interfacial segregation propensity have been neglected in Mg alloys. It remains to be established whether the notion of elastic strain minimization is the dominant effect in Mg alloyed with other elements, and how the chemical bonding effect, if any, would influence the solute segregation behavior.

In this work, based on observations made by atomic-resolution high-angle annular dark-field scanning transmission electron microscopy (HAADF-STEM), we report an unusual solute segregation phenomenon at CTBs in binary Mg–Bi, and Mg–Pb alloys, which is beyond the interpretation from the currently accepted notion. To uncover the physical origin, we also performed first-principles calculations. Our results suggest that chemical bonding is a key factor in determining solute segregation in CTBs. The unusual solute segregation, and the solute segregation behaviors reported previously, can be well rationalized by the chemical bonding effect.

## Results

**Large atoms segregated into compression sites of $\{10\bar{1}1\}$ CTBs.** Twins including $\{10\bar{1}1\}$ and $\{10\bar{1}2\}$ are ubiquitously observed in the deformed Mg–0.4Bi alloy, which is consistent with results in other Mg alloys[17,30]. The HAADF-STEM image in Fig. 1 shows a $\{10\bar{1}1\}$ CTB in a sample that was pre-compressed by 11% and then aged at 80 °C for 48 h. Bi and Mg have atomic radii of 1.70 and 1.60 Å, and atomic numbers of 83 and 12, respectively (Supplementary Table 1). Atomic columns rich in Bi are brighter in contrast in the HAADF-STEM image because the brightness of each column is approximately proportional to the square of atomic number[31,32]. Since extension and compression sites are distributed alternately in $\{10\bar{1}1\}$[13], Bi atoms having larger atomic size would be expected to segregate to the extension site of $\{10\bar{1}1\}$

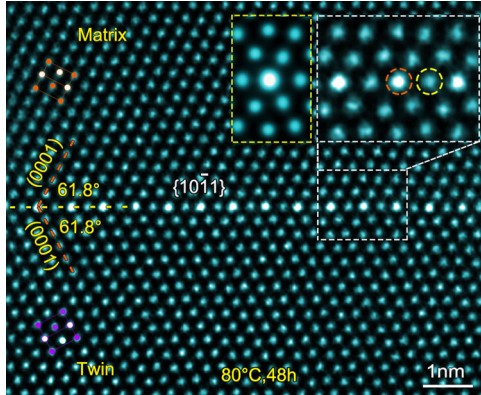

**Fig. 1 Periodic segregation of Bi atoms in $\{10\bar{1}1\}$ CTB.** HAADF-STEM image showing a $\{10\bar{1}1\}$ CTB in Mg–0.4Bi alloy. An enlarged CTB is shown in the upper-right inset, indicating Bi-rich columns at compression sites of $\{10\bar{1}1\}$ CTB. A simulated HAADF image, with the foil thickness of 75 nm (measured by PACBED), is inserted and enclosed by a yellow dotted-line rectangular frame. Periodical distribution of compression and extension sites in the CTB are marked by orange and yellow dash-line circles, respectively. Electron beam direction is parallel to $[1\bar{2}10]$.

to minimize elastic strain. Surprisingly, however, bright dots of Bi atoms are found at the compression sites (marked by an orange dash-line circle), as shown in the upper-right inset of Fig. 1. The Bi segregation is further confirmed in the energy-dispersive X-ray spectroscopy map (see Supplementary Fig. 1).

**Absence of evident segregation in $\{10\bar{1}2\}$ CTBs.** Figure 2a shows a HAADF-STEM image of a $\{10\bar{1}2\}$ CTB in Mg–0.4Bi alloy after 11% compression and aging at 160 °C for 1 h. Again, given the periodic distribution of extension and compression sites within $\{10\bar{1}2\}$[13], the larger Bi atoms are expected to occupy the extension sites to minimize the elastic strain energy. However, no bright dots are found in the $\{10\bar{1}2\}$. Even after samples are aged at 160 °C for 300 h, 80 °C for 48 h, or 80 °C for 200 h, (Fig. 2b–d), no apparent Bi segregation is found. There seems to be a lack of stability at the extension or compression sites of $\{10\bar{1}2\}$ for Bi segregation to occur. This anomalous segregation behavior at the $\{10\bar{1}1\}$ and $\{10\bar{1}2\}$ CTBs cannot be explained by the currently accepted notion that considers elastic strain minimization alone[13,21–24], implying that other factors should be taken into account when considering solute atom segregation in these CTBs.

**Segregation energies for solutes in CTBs.** Segregation energy derived from first-principles calculations is widely used to judge whether solute segregation in an interface is energetically favorable in Mg alloys and other metals[13,21,23,24,33,34]. Most previous studies only consider 100% occupancy of solute atoms in a single atomic column. In this work, we performed calculations with different solute occupancies at segregated sites of $\{10\bar{1}1\}$ and $\{10\bar{1}2\}$ CTBs, as shown in Fig. 3. Here, Gd is selected as a reference because its atomic radius is also larger and its segregation behavior can be well explained by elastic energy minimization[13]. Unexpectedly, the segregation energy curves of Bi at the compression and extension sites of $\{10\bar{1}1\}$ exhibit a transition (i.e., from energetically unfavorable to favorable) with decreasing solute Bi occupancies, which has not been reported before. With a 20% occupancy, the segregation energy value is the lowest among the five considered occupancies. In addition, the segregation energy curve of the compression site is always lower than that of the extension site for all occupancies, indicating that Bi segregation to the compression site is preferential when the Bi content is

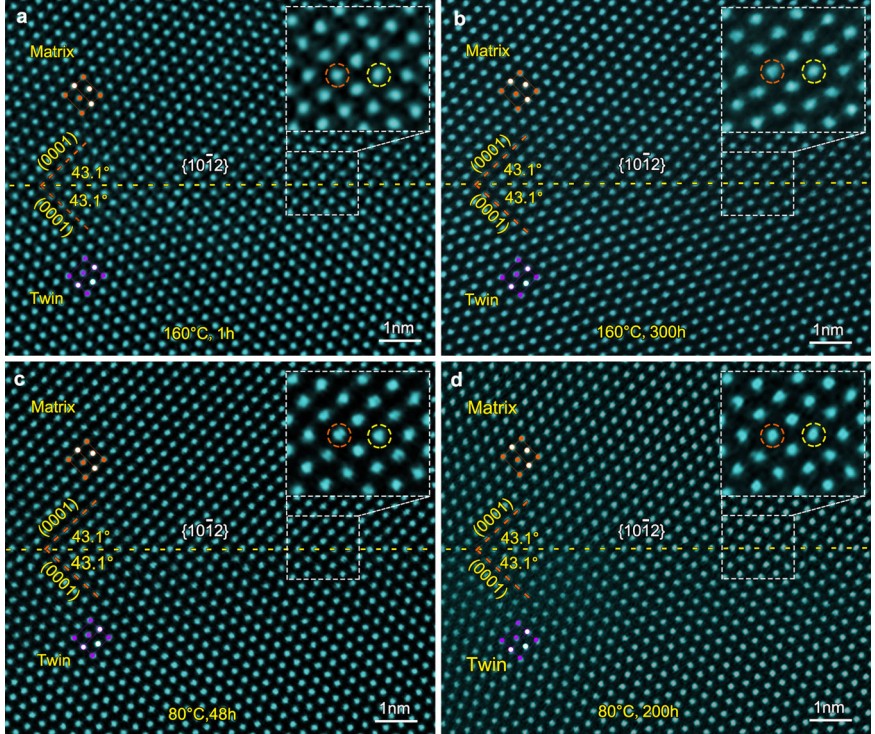

**Fig. 2 Absence of apparent segregation of Bi atoms in {10$\bar{1}$2} CTB.** HAADF-STEM images showing {10$\bar{1}$2} CTBs in Mg–0.4Bi alloy: **a** aged at 160 °C for 1 h, **b** 160 °C for 300 h, **c** 80 °C for 48 h, and **d** 80 °C for 200 h. Enlarged images of TBs are shown as insets in the upper-right, revealing that no Bi-rich columns are detected at {10$\bar{1}$2} CTBs. The alternating compression and extension sites of the CTB are marked by orange and yellow dash-line circles, respectively.

quantitatively low. Near-quantitative simulation of HAADF images (Supplementary Fig. 2) indicates that a solute occupancy of ~20 at% Bi in a Bi-rich column leads to image intensities matching well with those in the experimental image (Fig. 1). The occupancy of ~20 at% Bi at the compression site also agrees well with the strong segregation propensity predicted in Fig. 3a.

For {10$\bar{1}$2} CTBs, both compression and extension sites are calculated to be unfavorable for Bi segregation at all considered occupancies due to the positive segregation energy values, (Fig. 3b), and the extension sites are lower in energy than the compression sites. In contrast, the segregation energy values of Gd always are always negative at the extension site, and positive at the compression site, irrespective of the quantitative occupancy of Gd. Therefore, Gd atoms only segregate to the extension sites of the CTBs, as reported previously[13]. Note that previous first-principles calculations usually attribute the negative segregation energy to the elastic strain minimization. Here, although the common approach that is based on the reduction of elastic strain energy can interpret the Gd segregation at the extension sites, it cannot rationalize the segregation transition (Fig. 3a) and the unusual segregation phenomenon of Bi atoms (Figs. 1 and 2).

**Differential electron charge density analysis**. To reveal the physical origin of the unusual segregation behavior of Bi atoms, differential electron charge densities (DECD) of Bi occupying compression and extension sites of the CTBs were calculated. Here only the results of {10$\bar{1}$1} CTBs are shown in Fig. 4, and results for {10$\bar{1}$2} CTBs are available in Supplementary Fig. 3. For comparison, the DECDs of Gd and Zn in a {10$\bar{1}$1} CTB is also considered. Figure 4a–f shows DECD contour maps of Bi, Gd, and Zn atoms occupying compression and extension sites in {10$\bar{1}$1}, respectively. The charge density is higher around Bi and Zn atoms (area in red)

and lowers around Gd atom (area in blue) at both sites, which implies that Bi and Zn atoms attract valence electrons while Gd atom loses their valence electrons. This result also corresponds to the relationship predicted by electronegativity values of Bi, Gd, Zn, and Mg, which are 1.90, 1.20, 1.65, and 1.31, respectively. Note that Bi and Zn exhibit a common feature that the area with a high charge density is larger at the compression site than at the extension site, indicating that more valence electrons will be trapped when Bi or Zn segregate to the compression site. The trend in the case of Bi is revealed more clearly in the three-dimensional isosurface DECD pattern shown in Supplementary Fig. 4. Gd depletes more valence electrons when its segregation occurs at the extension site (Fig. 4c, d). Due to the fact that the more electrons transferred (trapped or depleted), the stronger the chemical bonding constructed, we speculate that the chemical bonding will affect the solute segregation behavior at the CTBs.

**Bader analysis**. To perform a quantitative analysis, Bader charge calculations were made and the values of Bi, Gd, and Zn occupying different sites of a CTB are given in Table 1. Bader charge analysis is a useful method to give a reliable approximation of the total electronic charge associated with an atom. Here, the Bader charges of solute atoms located in the Mg matrix are used as a reference. For solutes with a greater electronegativity than Mg (e.g., Bi and Zn) which attract electrons from the Mg matrix, a stronger chemical bond formed in a CTB would be indicated by a larger Bader charge value than that in the Mg matrix, whilst for less electronegative solutes (e.g., Gd) that donate electrons to the Mg matrix, stronger bonding is indicated by a smaller value. Bader analysis of the Bi atom located in the Mg matrix shows a gain of $2.21e$ from the surrounding Mg to the Bi solute, giving a net valence charge of $7.21e$. With these extra electrons, the Bader

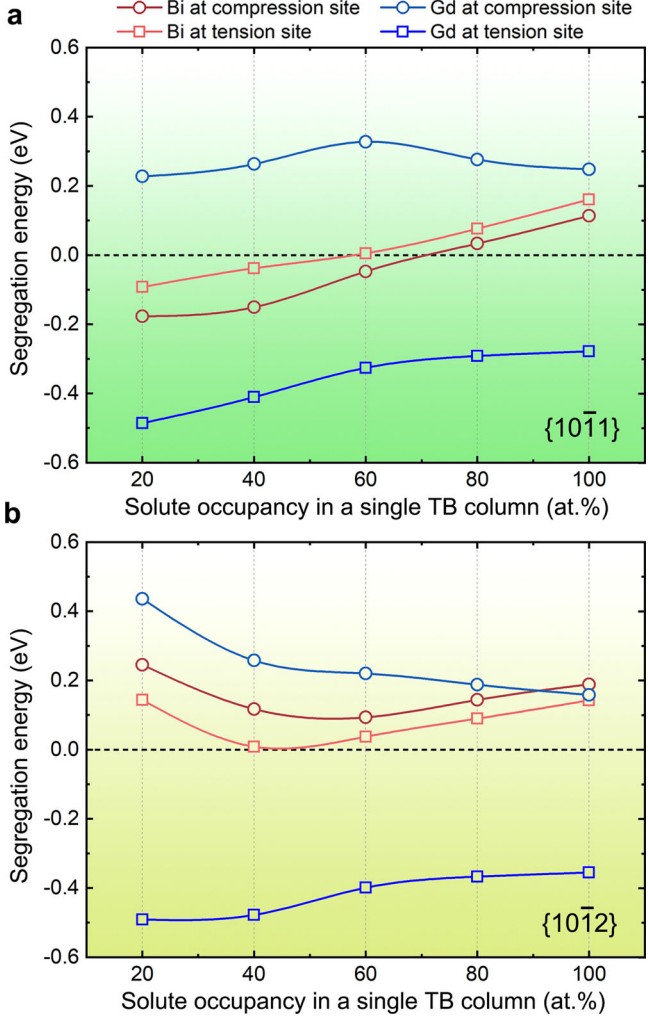

**Fig. 3 First-principles calculations of segregation energy.** Calculated solute segregation energies with different solute occupancies in a single atom column of **a** {10$\bar{1}$1} and **b** {10$\bar{1}$2} CTBs.

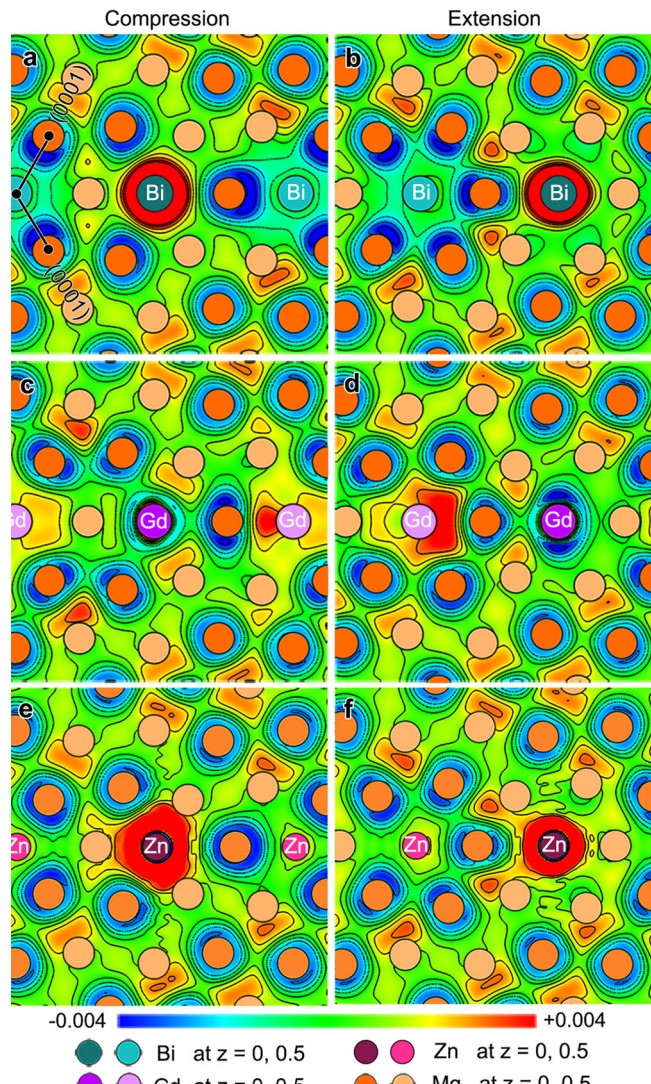

**Fig. 4 Calculated results of differential electron charge density (DECD).** (11$\bar{2}$0) plane contour maps of DECD of **a**, **b** Bi, **c**, **d** Gd, and **e**, **f** Zn segregated {10$\bar{1}$1} CTBs. The (11$\bar{2}$0) plane is close to the solute. The solute atoms in **a**, **c**, and **e** occupy compression sites in the CTB while those in **b**, **d**, and **f** are placed in extension sites. The black lines in **a** indicate (0001) planes of two crystals separated by {10$\bar{1}$1} CTB. The unit of scale bar is eV/Å$^3$.

volume of the solute Bi has grown to 51 Å$^3$ compared to the calculated Bader volume of 36 Å$^3$ of a Bi atom in Bi-metal. In the case of a 20% occupancy of Bi in the compression sites of {10$\bar{1}$1}, the Bader charge value (7.33$e$) is higher than that of Bi in the Mg matrix (7.21$e$) and that of Bi at the extension sites (6.98$e$), indicating stronger chemical bonding at the compression site. The increased charge on the Bi in the compression site also leads to a slight increase in its Bader volume to 52 Å$^3$. Due to the larger atomic radius of Bi, the local compression strain becomes intensified after Bi has segregated to the compression site. Thus, the stabilization effect from the stronger chemical bonding is considered to overcome the destabilizing effect from the more severe compression strain field, and this leads to favorable segregation energy of Bi at the compression site of {10$\bar{1}$1} for the case of 20% occupancy (Fig. 3) and the occurrence of the unusual segregation phenomenon (Fig. 1).

Similarly, other negative values in the segregation energy curves of Bi (Fig. 3) can also be explained by the factors of chemical bonding and elastic strain. In the case of Bi located at the extension site of {10$\bar{1}$2}, the segregation energy is calculated to be positive even though relocating a Bi atom from the matrix to the CTB expansion site can relieve strains in both the matrix and the CTB. The positive values can be attributed to the much

smaller Bader charge value at the expansion site than that in the Mg matrix (6.86$e$ vs. 7.21$e$). When Bi is located at the compression site, there is no evidence of stronger chemical bonds due to the similar Bader charge values (7.23$e$ vs. 7.21$e$) and no local strain relief, resulting in even higher segregation energy than that at the extension site. Therefore, the absence of Bi segregation in {10$\bar{1}$2} (Fig. 2) is attributed to the counterbalance between chemical bonding and strain field. In contrast, the segregation behaviors of Gd or Zn atoms are easier to predict. Zn segregation at the compression site of {10$\bar{1}$1} or {10$\bar{1}$2} gives rise to a higher Bader charge value than that in the Mg matrix, and the local compressive strain is also reduced due to the Zn segregation. The effects of the chemical bonding and elastic strain factors are beneficial for the Zn segregation at the compression site. Similarly, Gd segregation at the extension site of the CTBs is energetically preferred because of the satisfaction of both stronger chemical bonding and reduced local strains.

**Table 1 Bader charge values (e) of Bi, Gd, and Zn atom located in Mg lattice and occupying compression or extension sites in a CTB. The cases of 20% occupancy by Bi and Gd are also listed.**

| Solute | Mg matrix | Compression sites of {10$\bar{1}$1} | Extension sites of {10$\bar{1}$1} | Compression sites of {10$\bar{1}$2} | Extension sites of {10$\bar{1}$2} |
|---|---|---|---|---|---|
| Bi (100% occupancy) | 7.21[a] | 6.82 | 6.32 | 6.69 | 6.19 |
| Bi (20% occupancy) | – | 7.33 | 6.98 | 7.23 | 6.86 |
| Gd (100% occupancy) | 9.03[a] | 9.24 | 8.83 | 8.95 | 8.72 |
| Gd (20% occupancy) | – | 9.46 | 8.65 | 9.11 | 8.62 |
| Zn (100% occupancy) | 13.42[a] | 14.42 | 13.11 | 14.13 | 13.77 |
| Zn (20% occupancy) | – | 14.35 | 13.43 | 14.44 | 13.44 |

[a]Calculated from embedding a single solute atom in the Mg matrix.

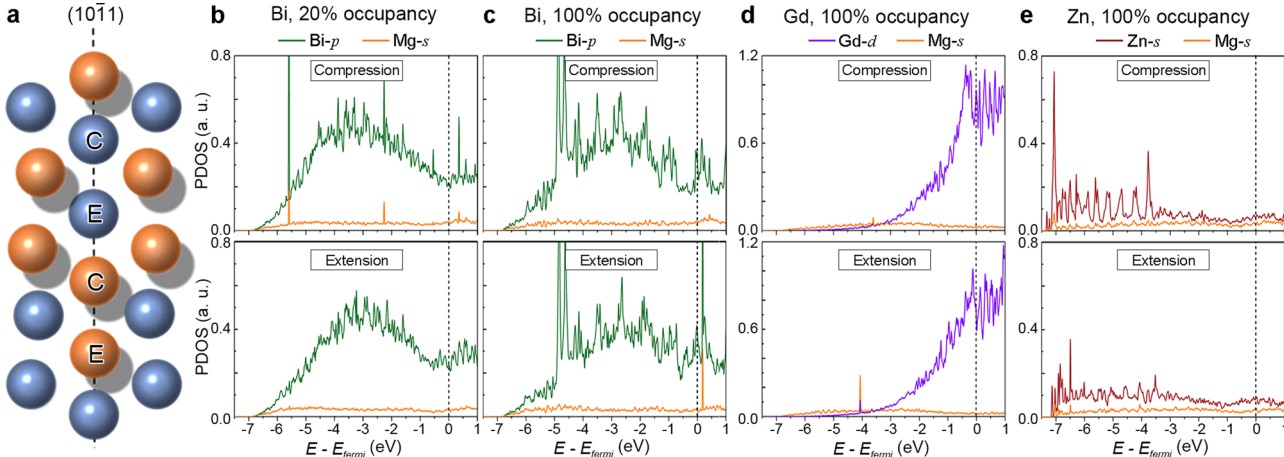

**Fig. 5 Site geometry and plots of the partial density of states (PDOS). a** Atomic arrangement of atoms at the {10$\bar{1}$1} CTB. Atoms labeled C show the compression site, and E denotes the expansion site. The orange and blue colors are used to denote atoms in different layers along [1$\bar{2}$10]. **b–e** PDOS for the cases where solute atoms segregated into the compression site or extension site of {10$\bar{1}$1} CTB: **b** Bi segregation with 20% occupancy, **c** Bi segregation with 100% occupancy, **d** Gd segregation with 100% occupancy, and **e** Zn segregation with 100% occupancy. The energies of states have been adjusted so that the Fermi level sits at 0 eV. The unit of PDOS in (**b–e**) is an arbitrary unit (a. u.).

**The density of states (DOS) analysis.** To verify the findings of chemical bonding in Bader analysis and to better understand the chemical bonding between the solute Bi, Gd, or Zn and its nearby Mg atoms, the electron density of states analysis was implemented for the cases where Bi (20 and 100% occupancy), Gd (100% occupancy), or Zn (100% occupancy) has segregated to the extension or compression site of {10$\bar{1}$1} CTB. Figure 5 plots the partial density of states (PDOS) of the solute atom and its closest Mg atom. In Fig. 5b, two intense hybridization peak pairs between Bi-$p$ and Mg-$s$ are found around −5.6 and −2.4 eV when a Bi atom has segregated to the compression site of {10$\bar{1}$1} with 20% occupancy, while no obvious bonding interaction is found for the case of the Bi has segregated to the extension site. When the occupancy of Bi at the compression site reaches 100%, no hybridization occurs, as shown in Fig. 5c. For the case of Bi segregation to the extension site with 100% occupancy, the hybridization peaks move to the energy higher than Fermi energy, which might lead to a more unstable electronic structure. Similar to the case of Bi segregation (20% occupancy) at the compression site, the presence of the hybridization peak pairs in Fig. 5d, e indicates strong electronic interactions, when Gd segregated to the extension site, or Zn into the compression site. The hybridization peak pairs under Fermi level in Fig. 5b, d would reduce the total energies of systems and stabilize the system structures. This also agrees well with the results of Bader's analysis.

**Discussion**
In the present work, Bi atoms having a larger atomic size than Mg occupy the compression site of {10$\bar{1}$1} (Fig. 1) and do not segregate to extension or compression sites in {10$\bar{1}$2} (Fig. 2). However, in previous experimental and theoretical studies, the segregation behavior of solute atoms at CTBs[13,14,21,22,27] and tilt GBs[18,33,35] in Mg alloys is interpreted only by the factor of atomic size effect by which a solute with a larger radius than Mg occupies the extension site while a solute with a smaller radius than Mg occupies the compression site of CTBs. It should be noted that the criterion based on the elastic strain field relaxation alone cannot predict the experimental observation in our work, as indicated by the strain energy values in Table 2 (the more positive value of the elastic strain energy contribution, the more local elastic strain would be released after the solute segregation). The strain energy along the twin boundary was calculated according to the method proposed by Huber et al.[36]. Systematic first-principles calculations suggest that the chemical bonding effect, combined with elastic strain minimization, can provide a more comprehensive view to the understanding of solute atom segregation to CTBs. Solute segregation at the compression site of the CTBs is energetically more favorable if the solute atom could reduce the local compressive strain (solute having a smaller radius than Mg), or attract more valence electrons to achieve a stronger chemical bonding (solute having a larger electronegativity value than Mg).

**Table 2 Elastic strain contributions to segregation energies.**

| Solute | Site | $E_{\text{seg}}^{\text{elastic}}$ (eV) |
|---|---|---|
| Bi | Compression site | −0.063 |
| | Extension site | 0.046 |
| Gd | Compression site | −0.142 |
| | Extension site | 0.103 |
| Zn | Compression site | 0.107 |
| | Extension site | −0.078 |

Cases of solute Bi, Gd, or Zn with 100% occupancy segregation to the compression or extension sites of (10$\bar{1}$1) CTB were calculated.

Contrarily, solute segregation at the extension site is more likely to occur when the solute atom has a larger metallic radius or has a lower electronegativity value. More specifically, the experimentally observed segregation pattern of Zn or Gd atoms satisfies both requirements to segregate to the compression or extension sites. But for Bi, there is a competition between the chemical bonding effect and local elastic strain minimization, leading to the unusual segregation phenomenon (Figs. 1 and 2).

This enhanced bonding comes about due to a combination of the coordination geometry of the sites and the valence electron orbital type. The expansion and compression sites not only have different sizes but different geometries, as seen in the {10$\bar{1}$1} CTBs in Fig. 5a. The compression site is surrounded by eleven Mg atoms, three in the same plane, with close to 120° spacing, then a set of three Mg atoms in the planes above and below, and finally Mg atoms directly above and below the compression site. From the enhanced charge transfer seen in the Bader analysis, this geometry would seem to be well suited to solutes whose valence electrons occupy $p$-type orbitals. The geometry of the expansion site in the {10$\bar{1}$1} CTBs is quite different, as are those of the compression and extension sites in the {10$\bar{1}$2} and {10$\bar{1}$3} CTBs (Supplementary Fig. 11).

To prove that our computational findings are not merely appropriate for solute Bi, Gd, or Zn, several other elements such as Pb, Tl, and In (Supplementary Table 1) that have the larger atomic size and similar electronegativity to Bi are also considered. Segregation energies at {10$\bar{1}$1} and {10$\bar{1}$2} CTBs with different occupancies of solutes are calculated (Supplementary Fig. 5). DECD images of Pb (Supplementary Fig. 6) and Bader charge values of Pb, Tl, and In segregation at CTBs (Supplementary Table 2) are also established. The calculation results reveal that these $p$-block elements do have a segregation behavior similar to Bi. Particularly, our experimental observations of segregation patterns in an Mg–Pb alloy (Supplementary Fig. 7) have confirmed our calculation results. Details are shown in the Supplemental Material.

We have also applied our findings to another type of CTB, which is {11$\bar{2}$1} that was detected in some RE-containing Mg alloys such as Mg–Y[37] and Mg–Gd[38]. Extension strain exists in the {11$\bar{2}$1} CTB[38], which favors RE atoms, larger than Mg, to segregate in the extension sites. We used the first-principles method to calculate the chemical bonding effect between the Y atoms located at {11$\bar{2}$1} CTBs and their surrounding Mg atoms. Calculation results suggest that the chemical bonding effect is relatively weak, leading to little segregation propensity, as indicated by the rather small value of the segregation energy (Supplementary Tables 3 and 4). Our experimental observations have verified such a prediction, i.e., no evidence of Y segregation is detected in {11$\bar{2}$1} CTBs of Mg–2 at% Y alloy that was precompressed by 5% and aged at 200 °C for 8 h (Supplementary Fig. 8a) or even 17 h (Supplementary Fig. 8b). These results suggest that our computation-based analysis can explain why no obvious Y segregation was observed experimentally.

In summary, Bi or Pb atoms that have atomic radii larger than that of Mg are experimentally found to segregate to the compression site in {10$\bar{1}$1} CTBs but not to the compression or extension site in {10$\bar{1}$2}. This unusual segregation phenomenon cannot be rationalized by the traditional criterion that involves exclusive consideration of the minimization of elastic strain energy. Our first-principles calculations suggest that it is the chemical bonding (coordination and solute electronic configuration) that dominates the solute segregation behavior in CTBs in these Mg alloys. The unusual segregation phenomenon observed in this work and the solute segregation behaviors at CTBs commonly detected in other Mg alloys can be all well explained by the chemical bonding effect. Our findings are expected to be applicable to CTBs and those coherent GBs in other HCP metals and perhaps even metals of other structures.

## Methods

**Materials and thermal treatments.** The materials used for experimental observations are Mg–0.4Bi and Mg–1.5Pb (at%) alloys. These alloys were prepared from high-purity Mg, Bi, and Pb metals by induction melting in a mild steel crucible at 760 °C under a mixed gas atmosphere ($SF_6 + CO_2$) and casting into a preheated steel mold. To avoid any local melting[39], the solution heat-treatment of the Mg–Bi alloy was performed at 320 °C for 24 h and then at 450 °C for 24 h while that of the Mg–Pb alloy at 300 °C for 24 h and then at 370 °C for 24 h, followed by cold water quenching. Small cylinder-shaped samples were uniaxially compressed to strains ranging from 3% to 20% at ambient temperature. After the compression test, these samples were aged at 80, 160, or 200 °C in oil baths.

**Electron microscopy characterization.** All the specimens for HAADF-STEM observations were ion-milled using Gatan PIPS 695 at −70 °C. HAADF-STEM characterization was conducted on a Cs-corrected FEI Titan G2 60–300 ChemiSTEM operated at 300 kV and equipped with a Super-X energy dispersive X-ray spectrometer. A 15 mrad convergence semi-angle and an inner-collection semi-angle of 57 mrad were used for HAADF imaging. The thickness of STEM foils was determined by the comparison between experimental position averaged convergent beam electron diffraction patterns and simulated patterns[40], and its value was used for quantitative simulation of HAADF-STEM images. Image simulations were performed using xHREM software package based on FFT-multislice and wave-optics[41]. Image simulation parameters of illumination conditions, such as convergent semi-angle, detector angle, defocus, and specimen thickness, were the same as experimental conditions. The atomic-scale HAADF-STEM images were processed by masking the corresponding diffraction patterns in the fast Fourier transforms of original images and then back transforming using Gatan Digital Micrograph.

**First-principles calculations.** Theoretical calculations involved in this work were completed using Vienna Ab initio Simulation Package[42] with the projected augmented wave method[43]. Generalized gradient approximation of Perdew–Burke–Ernzerhof[44] was applied to solve electron exchange and correlations. To simulate the {10$\bar{1}$1} and {10$\bar{1}$2} CTB structures in Mg alloys, four supercells with different atom numbers were constructed in our work, as shown in Supplementary Fig. 9. Each supercell contained two twin boundaries locating at the middle and bottom of the model. The distance between two twin boundaries was set far enough to avoid interactions. All the supercells were optimized by structural relaxations, where the energy and force convergences were $10^{-6}$ eV and $10^{-2}$ eV/Å. The cell parameter perpendicular to the twin boundary was free to vary, while the other two cell parameters were fixed to a multiple of the bulk Mg lattice constant to compensate for the effect caused by the artificially high number of twin boundaries in the simulation cell. For Brillouin zone sampling, the Monkhorst–Pack method was applied to generate $k$-point mesh. The supercells in Supplementary Fig. 9a–d were relaxed using $7 \times 29 \times 3$, $5 \times 4 \times 2$, $19 \times 25 \times 3$, and $5 \times 3 \times 2$ $k$-point meshes, respectively. The cut-off energy was set as 400 eV to offer enough degree of calculation accuracy. The segregation energies with different solute occupancies were calculated by placing different numbers of solute atoms in the atomic column of the compression or extension site, which were highlighted in Supplementary Fig. 9.

The segregation energy $E_{\text{seg}}$ was calculated using the expression as follows:

$$E_{\text{seg}} = \left\{ \left[ E_{\text{twin}}(\text{Mg}_{N-m}\text{X}_m) - E_{\text{twin}}(\text{Mg}_N) \right] - m \left[ E_{\text{bulk}}(\text{Mg}_{M-1}\text{X}) - E_{\text{bulk}}(\text{Mg}_M) \right] \right\}/m$$

(1)

where $E_{\text{twin}}(\text{Mg}_{N-m}\text{X}_m)$ is the total energy of supercell containing solute segregated CTBs, $E_{\text{twin}}(\text{Mg}_N)$ is the total energy of CTB supercell without solute atoms, $N$, $N - m$, and $m$ is the number of atoms in the supercell, the number of Mg atoms and the number of solute atoms, respectively. $E_{\text{bulk}}(\text{Mg}_{M-1}\text{X})$ is the total energy of a supercell representing where a solute atom dissolves in the Mg matrix. $E_{\text{bulk}}(\text{Mg}_M)$ is the total energy of a supercell in a perfect solute-free Mg lattice. When determining the segregation energy at a specific solute occupancy, we have considered different distributions of solute atoms in a single atomic column and found that the differences of segregation energies between these

distributions are negligible. Consequently, the segregation energies from the situation where solute atoms distribute uniformly were selected and displayed in our figures.

Differential electron charge density $\rho_{DECD}$ of a fully relaxed supercell in this work is given by the electron charge with static self-converged electronic relaxation subtracting the charge density with only one step electronic relaxation:

$$\rho_{DECD}\left(Mg_{N-m}X_m\right) = \rho_{inter}\left(Mg_{N-m}X_m\right) - \rho_{non-inter}\left(Mg_{N-m}X_m\right) \quad (2)$$

where $\rho_{inter}$ is the charge density of supercell after completely converged electronic relaxations, and $\rho_{non-inter}$ is the reference charge density (or non-interaction charge density) obtained by applying only one step of electronic relaxation to the fully relaxed supercell. $N-m$ and $m$ are the number of Mg atoms and solute atoms in the input supercell, respectively. The DECD is widely utilized in other simulation studies regarding diffusion or solute-defect interaction to investigate the interactions between Mg atoms and solute atoms[45,46].

Bader analysis[47] applied in this work was implemented using the code from Henkelman et al.[48] to obtain reliable approximations of the total electronic charge of a solute atom.

The schematic diagrams showing atomic models and charge densities involved in this work were all created by VESTA packages[49].

## Data availability

All relevant data supporting the findings of this study are contained in the paper and its Supplementary Information files. All other relevant data are available from the corresponding authors on request.

## Code availability

The DFT calculations were made by the Vienna Ab initio Simulation Package (VASP) that is from the University of Vienna.

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

## Acknowledgements

This research was financially supported by the National Natural Science Foundation of China (51771036, 51131009, and 51421001), the National Key Research and

Development Program of China (2016YFB0700402), and the Graduate Research and Innovation Foundation of Chongqing, China (Grant no. CYB17004) and the Australian Research Council. C.H. would like to thank A.P.Z. for her assistance and suggestions on the preparation of TEM samples. The authors also gratefully acknowledge the access to the facilities of the Electron Microscopy Center in Chongqing University. This work was supported by computational resources provided by the Australian Government through PAWSEY and NCI under the National Computational Merit Allocation Scheme.

## Author contributions

C.H., H.W.C., and J.F.N. conceived the idea and designed the experiments. C.H. performed the HAADF-STEM and analyzed the results. Z.Q.L. and N.W. carried out the first-principles calculations. H.W.C. did HAADF-STEM simulation. All the authors discussed and contributed to the paper.

## Competing interests

The authors declare no competing interests.
