## [Peer Review File · Nature Communications]

REVIEWER COMMENTS

Reviewer #1 (Remarks to the Author):

The authors have presented a very interesting study of the segregation of Bi to coherent twin boundaries in Mg (and comparisons with more frequently studied Al, Zn, and Gd as well as other species Pb, Th, and In). The results are indeed very interesting, because there are details of the segregation behavior which go against conventional wisdom based solely on arguments related to solute size and interfacial site size. However, the work is both undersold and oversold throughout the paper. It is recommended that the following revisions be made.

First of all, there is no reason to expect that the results would be restricted to twin boundaries in Mg or even hexagonal metals and alloys. The concluding statements in the abstract and elsewhere should be broadened to highlight the applicability of the concepts revealed to a broader class of materials, at least metals and alloys and especially coherent grain boundaries, but as already suggested by the authors, we should expect the concepts to apply to general grain boundaries as well.

On the other hand, every serious student of materials science will know that there is both a mechanical and a chemical contribution to the free enthalpy of segregation (to the boundary, in general, and the individual boundary site, in specific). Thus, it actually comes as no surprise that one must consider both contributions when evaluating the propensity to segregate to an individual site. It is not correct to imply that researchers would traditionally only consider strain energy criteria.

In short, because the journal Nature Communications is directed at a general scientific audience (not only materials science), the paper should be better contextualized within the framework of the actual traditional understanding of interfacial segregation. Even a Wikipedia entry on the subject goes beyond the most simplistic McLean-Langmuir concept, which indeed relied upon elastic strain energy contributions alone, to explain the propensity to segregate to an interface in general.

Even the formation free energy of the solute atom in the matrix has both mechanical and chemical contributions. To say otherwise would deny the most basic of empirical alloy theories, Hume-Rothery's rules governing significant solubility. Indeed, only 1 of his 4 basic criteria have to do with strain energy (the size factor), whereas 2 of 4 concern electronic structure (valence and electromotive series). Thus, the findings of the authors are a validation of concepts which have been employed by materials scientists for decades. The authors would do well to acknowledge this fact rather than repeatedly state how surprising their results are. They are primarily surprising to the authors because they found that the first-order approximation worked for their prior observations. Perhaps that should be labeled as the real surprise.

Note that the early work of Seah, M. P., & Hondros, E. D. (1973). Grain boundary segregation. Proceedings of the Royal Society of London. A. Mathematical and Physical Sciences, 335 (1601), 191-212 highlighted the strong correlation between propensity to segregate and solid solubility within the surrounding matrix. It would be very interesting to learn the author's opinions regarding their

observations regarding possible correlations with the relative solubility of the species considered. (Bi is not very soluble, whereas Al, Gd, In, Th, and Zn are highly soluble in Mg.

See the recent work of Huber, L., Grabowski, B., Militzer, M., Neugebauer, J., & Rottler, J. (2017). Ab initio modelling of solute segregation energies to a generic grain boundary. *Acta Materialia*, 132, 138-148. In this work, they show that segregation is indeed governed by mechanical and chemical contributions computed from first principles.

Finally, since the authors have gone all the way to the point of performing Bader analysis, they will appreciate the fact that even the distinction between chemical and mechanical contributions is artificial. All of these effects have to do with the details of electronic structure. Bader himself introduced the means by which the individual atom could be parsed from the molecule or crystal in which it is embedded, based upon the electronic structure. Perhaps the authors could recalculate the elastic strain energy contribution based upon the size of the solute atom (when it is embedded within the matrix, as opposed to being within its own crystal structure). They may find that the change in size of the Bi atom is larger than initially anticipated, when it is dissolved in Mg. Thus, they may find an additional explanation which is consistent with what they have already written. Alternatively, if they perform the analysis of Huber et al., they may be able to better parse the mechanical and chemical contributions.

The authors may contest that there is not space to address the concerns raised. However, I believe they can find space if they remove the abundant redundancy in current manuscript which repeats the notion that their observations are surprising over and over again.

Detailed comments and questions:

Are there only three types of coherent twins within Mg and its alloys? Have old reports by the likes of Reed-Hill and C.S. Roberts of other odd twin types been refuted in recent times? Note that Nicole Stanford recently revealed a novel twin-type in Mg-Y alloys. Thus, at least this twin type should be added.

There very minor typographical errors throughout, which should be checked again.

Reviewer #2 (Remarks to the Author):

This manuscript focuses attention on the factors that determine segregation energies to some coherent twin boundaries in Mg. The authors prove experimentally that rules of thumb based on atomic size are inadequate in predicting the observed segregation behavior for a number of elements. They authors then go on to argue that the nature of chemical bonding is really the driving force, and that this chemical bonding can sometimes be in conflict with the strain effects.

I like this paper overall, and think that it will ultimately be worthy of publication in Nature Communications. However, I do have a few points that I would like the authors to consider.

First, the Bader analysis indicates that more strong bonding has occurred, but in some sense, it doesn't result in the bonding (a phrase used in the manuscript). That is the Bader analysis isn't causal but reflective. So from the analysis, we know that there is stronger bonding, but we do not know why. In this respect, I would have liked to see the authors analyze the electronic structure a bit more carefully. Plotting local density of states in order to determine which Bi states are bonding with which Mg states, for example, might be very insightful, and may lead to a predictive capability for which atoms will be segregate to which sites, without the need to compute the segregation energies explicitly.

The remaining points are more minor. The description of the computation of the differential electron charge density is a bit unclear. What is the starting point for the single iteration result that serves as the benchmark? Is this the overlap of atomic charges? What is the physics that makes this a good and reproducible choice. Also, the relaxation of the supercell is mentioned, but it is unclear to me which axes of the supercells were allowed to relax? If it was all axes, then I am a bit confused, as the lattice parameters in the plane of the tilt boundary should be fixed. (The tilt boundaries may necessarily change the lattice parameters, but this is an artifact of their extreme high density in the periodic supercell approach.) Also, how was the lattice parameter for the alloys fixed? Did the authors use a bulk calculation of a random solid solution alloy to determined these parameters?

I think that the authors need to address these issues before publication of their manuscript.

Reviewer #3 (Remarks to the Author):

The authors report a combined experimental-computational study on the segregation behaviour of solute atoms to twin boundaries in binary Mg-Bi and Mg-Pb alloys. Using HAADF-HRSTEM they observed segregation of these solute atoms to compression sites in 10-11 twin boundaries instead of to extension sites. These results are unexpected and counter-intuitive as segregation towards extension sites would be expected from the atomic radii of the solute atoms. In order to understand this effect, the authors have performed ab initio calculations considering not only the minimization of elastic strain but also the chemical bonding character in terms of charge transfer. These computational results enabled the authors to conclude that not only the atomic size but also the

electronic charge transfer of the atomic bonds determines the segregation sites in twin boundaries in Mg alloys.

The authors compare their results with data published before in literature and could give indications and expand their interpretation also to these data from literature.

I very much enjoyed reading the manuscript. It is well structured and the results and conclusions are exciting and expand the current view on the mechanisms of the interactions of solute atoms with structural defects.

The TEM micrographs are of high quality and the image simulations convince. There is one small question regarding the appearance of the 10-11 twin boundary in Mg-Pb (Suppl. Fig. 7): it appears that a faceting of the twin boundary is present. Do the authors think this could be resulting from the Pb segregation to minimize the elastic strain energy induced by Pb segregation or do they interpret this differently?

Response to Reviewers' Comments

We would like to thank the reviewers for their careful reading of and constructive comments on our paper. We have revised our manuscript accordingly. We have also added eight additional references, one additional figure, and one additional table in the revised manuscript, together with three additional figures and two tables in the revised supplementary information. All revisions are highlighted with yellow background in the manuscript and the supplementary information. We also added Dr. Nick Wilson as a co-author, as he made contributions to some of the additional DFT work in the revision. We hope that the revised version is now acceptable for publication in Nature communications.

Below are our responses (in blue color) to their comments.

Reviewer #1

1. First of all, there is no reason to expect that the results would be restricted to twin boundaries in Mg or even hexagonal metals and alloys. The concluding statements in the abstract and elsewhere should be broadened to highlight the applicability of the concepts revealed to a broader class of materials, at least metals and alloys and especially coherent grain boundaries, but as already suggested by the authors, we should expect the concepts to apply to general grain boundaries as well.

In the abstract of the revised manuscript, “They are likely to provide insights into interface boundaries in other hexagonal metals and alloys.” is now changed into “They are likely to provide insights into interface boundaries in other metals and alloys of different structure.” We also added the following sentence to the end of the concluding remarks in the revision:

“Our findings are expected to be applicable to CTBs and those coherent grain boundaries in other HCP metals and perhaps even metals of other structures.”

In the revision, we have applied our finding (solute segregation dominated by strong chemical bonding effect) to successfully interpret the experimental results of $\{11\bar{2}1\}$ CTBs. This part was added as a new paragraph at the end of Discussion (see the details in the response to Comment 6).

2. On the other hand, every serious student of materials science will know that there is both a mechanical and a chemical contribution to the free enthalpy of segregation (to the boundary, in general, and the individual boundary site, in specific). Thus, it actually comes as no surprise that one must consider both contributions when evaluating the propensity to segregate to an individual site. It is not correct to imply that researchers would traditionally only consider strain energy criteria. In short, because the journal Nature Communications is directed at a general scientific audience (not only materials science), the paper should be better contextualized within the framework of the actual traditional understanding of interfacial segregation. Even a Wikipedia entry on the subject goes beyond the most simplistic McLean-Langmuir concept, which indeed relied upon elastic strain energy contributions alone, to explain the propensity to segregate to an interface in general.

Indeed, contributions from strain energy and chemical bonding govern the solute segregation into interface boundaries of the metallic and non-metallic materials. However, in Mg alloys, all of the experimental and computational studies have correlated the interfacial segregation of solute atoms with the elastic strain energy due to the atomic size mismatch between the metal and the solute. The effect of chemical bonding on the solute segregation behaviours at coherent twin boundaries (CTBs) of Mg alloys seems to be neglected in previous studies. It is for this reason that we stated that “the currently accepted notion only considers elastic strain minimization to explain the solute segregation into CTBs of Mg alloys” in the original manuscript. To make it clearer, we have changed the description in the end of Paragraph 2 of the Introduction in the revision, and it now reads:

“This can be understood by considering a large solute in a Mg lattice, where the solute will displace the surrounding Mg atoms away from their normal lattice positions, creating a local strain. If the solute relocates to an extension site in a CTB, then it is still bonded with Mg, however strain in the lattice is removed, and the strain in the CTB extension site is reduced. To date, this notion of the elastic strain minimization has successfully explained all the experimentally observed solute segregation behaviours in CTBs in Mg alloyed with a rare-earth (RE), Zn, or Ag, and Mg alloyed with multiple elements, such as RE and Zn, or RE and Ag^{13,14,25-27}. Although chemical bonding is reported as another factor that affects solute segregation in ceramic and semiconductor materials^{28,29}, its effects on interfacial segregation propensity has been neglected in Mg alloys. It remains to be established whether the notion of elastic strain minimization is the dominant effect in Mg alloyed with other elements, and how the chemical bonding effect, if any, would influence the solute segregation behaviour.”

Also, we have changed the sentence in the Abstract, “We propose that such segregation is driven by chemical bonding rather than elastic strain minimization.”, into “We propose that such segregation is dominated by chemical bonding (coordination and solute electronic configuration) rather than elastic strain minimization.”.

The sentence in the concluding remarks, “Our first-principles calculations suggest that it is the chemical bonding that influences the solute segregation behavior in CTBs in these Mg alloys.”, into “Our first-principles calculations suggest that it is the chemical bonding (coordination and solute electronic configuration) that dominates the solute segregation behavior in CTBs in these Mg alloys.”.

3. Note that the early work of Seah, M. P., & Hondros, E. D. (1973). Grain boundary segregation. Proceedings of the Royal Society of London. A. Mathematical and Physical Sciences, 335 (1601), 191-212 highlighted the strong correlation between propensity to segregate and solid solubility within the surrounding matrix. It would be very interesting to learn the author’s opinions regarding their observations regarding possible correlations with the relative solubility of the species considered. (Bi is not very soluble, whereas Al, Gd, In, Th, and Zn are highly soluble in Mg).

In the work of Seah and Hondros^{R1}, the empirical correlation between propensity to segregate and solid solubility in the matrix is that the segregating tendency of an element is inversely related to its limit of solid solubility at the ageing temperature. However, this empirical

correlation neglects structural differences of various grain boundaries (e.g. different twin boundaries). In our work, we found that Bi and Pb exhibit a similar segregation behaviour, i.e., atoms of Bi or Pb segregate to compression sites of $\{10\bar{1}1\}$ fully coherent twin boundary (CTB) and do not segregate to the extension or compression site of $\{10\bar{1}2\}$ CTB, even though Bi is not very soluble ($<0.01\text{at\%}$ at $200\text{ }^\circ\text{C}$) but Pb is highly soluble ($\sim 0.93\text{at\%}$ at $200\text{ }^\circ\text{C}$) in Mg. Therefore, we conclude that the cases of Mg–Bi and Mg–Pb alloys do not follow the empirical correlation between the solubility of solute and its propensity to segregate.

4. Finally, since the authors have gone all the way to the point of performing Bader analysis, they will appreciate the fact that even the distinction between chemical and mechanical contributions is artificial. All of these effects have to do with the details of electronic structure. Bader himself introduced the means by which the individual atom could be parsed from the molecule or crystal in which it is embedded, based upon the electronic structure. Perhaps the authors could recalculate the elastic strain energy contribution based upon the size of the solute atom (when it is embedded within the matrix, as opposed to being within its own crystal structure). They may find that the change in size of the Bi atom is larger than initially anticipated, when it is dissolved in Mg. Thus, they may find an additional explanation which is consistent with what they have already written. Alternatively, if they perform the analysis of Huber et al., they may be able to better parse the mechanical and chemical contributions.

We agree with the reviewer that the distinction between chemical and mechanical contributions is an artificial partitioning of the energies from the electronic structure and thank them for raising the issue of the changing size of the Bi atom when embedded in the Mg matrix. There is no unique definition for the size of an atom in a solid; different schemes for determining atomic size will produce different values. Bader analysis provides a method for partitioning the electron charge density between atoms based on the topology of the charge density, and from this atomic size calculated. In pure Bi metal the Bader volume is calculated to be 36 \AA^3 , but when a Bi atom is placed in the Mg lattice there is a charge transfer of around $2.2|e|$ from the surrounding Mg atoms to the Bi atom which results in an expansion of its Bader volume of the Bi solute to 51 \AA^3 . So, indeed, the Bi atom, from the point of view of Bader analysis, has grown in size when dissolved in Mg. When placed at the twin boundary compression site, there is a slight increase in size to 52 \AA^3 . We have added these details to the Bader analysis section of the manuscript and it now reads:

“To perform a quantitative analysis, Bader charge calculations were made and the values of Bi, Gd and Zn occupying different sites of a CTB are given in Table 1. Bader charge analysis is a useful method to give a reliable approximation of the total electronic charge associated with an atom. Here, the Bader charges of solute atoms located in the Mg matrix are used as a reference. For solutes with a greater electronegativity than Mg (e.g. Bi and Zn) which attract electrons from the Mg matrix, a stronger chemical bond formed in a CTB would be indicated by a larger Bader charge value than that in the Mg matrix, whilst for less electronegative solutes (e.g. Gd) that donate electrons to the Mg matrix, stronger bonding is indicated by a smaller value. Bader analysis of Bi atom located in the Mg matrix shows a gain of $2.21\text{ }e$ from the surrounding Mg to the Bi solute, giving a net valence charge of $7.21\text{ }e$. With these extra electrons, the Bader volume of the solute Bi has grown to 51 \AA^3 compared to the calculated Bader volume of 36 \AA^3 of a Bi atom in Bi metal. In the case of a 20% occupancy of Bi in the compression site of $\{10\bar{1}1\}$, the Bader charge

value (7.33 e) is higher than that of Bi in the Mg matrix (7.21 e) and that of Bi at the extension site (6.98 e), indicating stronger chemical bonding at the compression site. The increased charge on the Bi in the compression site also leads to a slight increase in its Bader volume to 52 \AA^3 . Due to the larger atomic radius of Bi, the local compression strain becomes intensified after Bi has segregated to the compression site. Thus, the stabilization effect from the stronger chemical bonding is considered to overcome the destabilizing effect from the more severe compression strain field, and this leads to a favourable segregation energy of Bi at the compression site of $\{10\bar{1}1\}$ for the case of 20% occupancy (Fig. 3) and the occurrence of the unusual segregation phenomenon (Fig. 1).”

We have recalculated the elastic strain energy contribution according to the calculation method proposed by Huber *et al.*³⁶. As shown in Table R1, the prediction of our calculated strain energy values is consistent with what we have stated in our manuscript, i.e., the over-sized solute, Bi or Gd, releases the local elastic strain after it has segregated to the extension site of CTB and under-sized solute such as Zn reduces the local elastic strain when segregated to the compression site. In the revision, we have added a table that summarizes the related elastic strain energy contribution. The corresponding description in the first paragraph of Discussion now reads:

“It should be noted that the criterion based on the elastic strain field relaxation alone cannot predict the observation in our work, as indicated by the strain energy values in Table 2 (the more positive value of the elastic strain energy contribution, the more local elastic strain would be released after the solute segregation). The strain energy along the twin boundary was calculated according to the method proposed by Huber *et al.*³⁶.”

According to the calculation method of Huber *et al.*³⁶, we have also calculated the contributions of strain energy and chemical bonding. However, the calculated value of $E_{seg}^{elast+bond}$ cannot interpret the experimental results, such as solute atom segregation behaviors at CTBs in Mg–Zn and Mg–Bi alloys. This might be caused by the complex electronic structure of solutes, as has been mentioned in the work of Huber *et al.*³⁶. For example, the solute Pb and Bi have a similar and complex electron configuration (Pb, $[\text{Hg}]6p^2$, Bi, $[\text{Hg}]6p^3$). The segregation case regarding the former solute Pb showed relatively larger calculation error³⁶, and the latter Bi case utilizing the same method in the present work also failed to predict its segregation behavior (Table R1). Optimization of the calculation method is out of the scope in the present work and more work is required in the future. Details of the related calculation results are as follows:

To make sure that we understand the calculation approach properly, we repeated the work about the solute segregations (Mg and Pb) in the tilt boundary $\Sigma 5$ in Al and made a comparison between our recalculation result and their published results, as shown in the Fig. R1. Our recalculation results were close to theirs, which indicates our following calculations regarding the $(10\bar{1}1)$ CTB in Mg alloys are reliable. Note that this approach³⁶ can only consider the situations with 100% occupancy. Herein, we also only calculated the cases of solute segregation with 100% occupancy. The calculated elastic strain energy contributions and chemical bonding contributions to segregation energies are listed in Table R1. Note that the meaning of $E_{seg}^{elast+bond}$ here is different from our previous definition of E_{seg} . The more positive value of $E_{seg}^{elast+bond}$ stands for the stronger segregation propensity. In Table R1, it is found that the related elastic strain energy contributions are consistent with what we stated in our manuscript, that is, the

oversized solute, Bi or Gd, releases the local elastic strain when segregating into the extension site of CTB and undersized solute such as Zn reduces the local elastic strain when segregating into the compression site. However, the calculated value of $E_{seg}^{elast+bond}$ cannot interpret the experimental observations, i. e., solute Zn is predicted to segregate into extension site rather than compression site of CTB according to the values 0.169 eV vs -0.242 eV. Also, the predicted segregation for Bi from the approach is not consistent with the experimental result shown in Fig. 1.

Table R1. Segregation energies and their elastic strain contributions and chemical bonding contributions. Cases of solute Bi, Gd and Zn with 100% occupancy segregating into the compression sites or extension sites of $(10\bar{1}1)$ CTB were calculated.

Solute	Site	$E_{seg}^{elastic}$ (eV)	$E_{seg}^{bonding}$ (eV)	$E_{seg}^{elast+bond}$ (eV)
Bi	Compression site	-0.063	-0.278	-0.341
	Extension site	0.046	0.197	0.243
Gd	Compression site	-0.142	-0.307	-0.449
	Extension site	0.103	0.217	0.320
Zn	Compression site	0.107	-0.349	-0.242
	Extension site	-0.078	0.247	0.169

Figure R1. Comparison between the present recalculation results and the results reported by Huber et al. Recalculations of the results established in the calculation work of Huber et al. regarding solute segregation (Mg and Pb) on tilt boundary $\Sigma 5$ in Al. (a) Recalculation results based on first principles. (b) Results in the work of Huber *et al.*³⁶.

- Are there only three types of coherent twins within Mg and its alloys? Have old reports by the likes of Reed-Hill and C.S. Roberts of other odd twin types been refuted in recent times?

No, there are actually four types of primary twins reported to occur in Mg and its alloys, including $\{10\bar{1}1\}$, $\{10\bar{1}2\}$, $\{10\bar{1}3\}$ and $\{11\bar{2}1\}$ twins. The first three types are commonly observed in pure Mg and most Mg alloys and the last type is experimentally observed only in those RE-containing Mg alloys, such as Mg–Y³⁷ and Mg–Gd alloys³⁸.

Yes, the existence of the odd twin type in Mg (i.e. $\{30\bar{3}4\}$ coherent twins) has been refuted. Using optical microscopic techniques, Reed-Hill and Roberts^{R2} indeed reported the observation of so-called $\{30\bar{3}4\}$ coherent twins in Mg in the 1950s. However, in a later report^{R3}, using the high-resolution electron microscope, they found that $\{30\bar{3}4\}$ twins are actually formed by $\{10\bar{1}1\}$ - $\{10\bar{1}2\}$ double-twin mechanism, which has also been confirmed in recent studies using advanced electron backscattered diffraction (EBSD) or TEM (transmission electron microscopy) techniques^{R4,R5}. That is to say, the so-called $\{30\bar{3}4\}$ CTB is actually of secondary $\{10\bar{1}1\}$ - $\{10\bar{1}2\}$ type whose boundary is thought to be highly incoherent (large angle of misfit)^{R6}. Since our paper mainly reports solute segregation behaviors within those fully coherent twin boundaries, the secondary $\{10\bar{1}1\}$ - $\{10\bar{1}2\}$ twin boundary is not contained in the revised manuscript.

6. Note that Nicole Stanford recently revealed a novel twin-type in Mg-Y alloys. Thus, at least this twin type should be added.

The less common twin-type in Mg-Y alloys recently revealed by Nicole Stanford is the $\{11\bar{2}1\}$ twin. To date, $\{11\bar{2}1\}$ twins were observed only in those RE-containing Mg alloys, such as Mg-Y³⁷ and Mg-Gd alloys³⁸. In the Mg-0.3at%Bi and Mg-2.5at%Pb alloys, $\{11\bar{2}1\}$ twins were not detected. We did observe $\{11\bar{2}1\}$ twins in the Mg-2at%Y alloy. However, no solute segregation is observed in $\{11\bar{2}1\}$ CTBs of Mg-2at%Y alloy that was pre-compressed by 5% and then aged at 200 °C for 8 hours (Supplementary Figure 8a) or even 17 hours (Supplementary Figure 8b). Note that Y segregation was detected in $\{10\bar{1}2\}$ CTBs and precipitates were formed in the Mg matrix, after ageing treatment for 8 and 17 hours at 200 °C (not shown). To find out whether our conclusions are applicable to $\{11\bar{2}1\}$ twins in Mg alloys, first-principles calculations were also carried out and the results are shown in the following. Supplementary Figure 10 is the atomic diagram to show the input $\{11\bar{2}1\}$ twin supercell. Three possible segregation sites Site 1-3 were considered. The segregation energy values of these sites are listed in Supplementary Table 3. All the values are rather small in scale, indicating that Y atoms have little segregation propensity at $\{11\bar{2}1\}$ CTB, since most obvious solute segregation phenomena at CTBs of Mg alloys correspond to a much more negative segregation energy value^{13,23}. Bader charge analysis was also further carried out, and the main results are listed in Supplementary Table 4. Bader charge values of Y atoms segregating into the three sites are all similar to that of Y substituting in the Mg matrix, which suggests that the chemical bonding effect is relatively weak. These results imply that our conclusion (solute segregation dominated by strong chemical bonding effect) is applicable to explain why no obvious Y segregation was observed experimentally. In the revision, we have added the results of $\{11\bar{2}1\}$ CTBs as a new paragraph at the end of Discussion, and it reads:

“We have also applied our findings to another type of coherent twin boundary, which is $\{11\bar{2}1\}$ that was detected in some RE-containing Mg alloys such as Mg-Y³⁷ and Mg-Gd³⁸. Extension strain exists in the $\{11\bar{2}1\}$ CTB³⁸, which favors RE atoms, larger than Mg, to segregate in the extension sites. We used first-principles method to calculate the chemical bonding effect between the Y atoms located at $\{11\bar{2}1\}$ CTBs and their surrounding Mg atoms. Calculation results suggest that the chemical bonding effect is relatively weak, leading to little segregation propensity, as indicated by the rather small value of the segregation energy (Supplementary Tables 3 and 4). Our experimental observations have

verified such a prediction, i.e., no evident of Y segregation is detected in $\{11\bar{2}1\}$ CTBs of Mg-2at%Y alloy that was pre-compressed by 5% and aged at 200 °C for 8 hours (Supplementary Fig. 8a) or even 17 hours (Supplementary Fig. 8b). These results suggest that our computation-based analysis can explain why no obvious Y segregation was observed experimentally.”

7. There very minor typographical errors throughout, which should be checked again.

We have thoroughly proof read our manuscript and corrected all typographical errors that we could find. All corrections are highlighted by the yellow background colour.

Reviewer #2

1. So from the analysis, we know that there is stronger bonding, but we do not know why. In this respect, I would have liked to see the authors analyse the electronic structure a bit more carefully. Plotting local density of states in order to determine which Bi states are bonding with which Mg states, for example, might be very insightful, and may lead to a predictive capability for which atoms will be segregate to which sites, without the need to compute the segregation energies explicitly.

We have calculated the electron density of states of cases where Bi (20% and 100% occupancy), Gd (100% occupancy), or Zn (100% occupancy) has segregated into the compression or extension sites of $\{10\bar{1}1\}$ CTB, as shown in Fig. 5. Figures 5b and 5c show the partial density of states from a Bi atom and its closest Mg atom, when the Bi atoms have segregated to the compression or extension sites of the $\{10\bar{1}1\}$ CTB with 20% and 100% occupancy. In Fig. 5b, the presence of intensive hybridization peak pairs between Bi-*p* and Mg-*s* orbitals around -5.6 eV and -2.4 eV indicates strong chemical bonding between the segregated Bi and nearby Mg atoms, leading to the observed segregation at the compression sites (Fig. 1). Similarly, the presence of hybridization peak pairs in Figs. 5d indicates strong electronic interactions when Gd segregated to the extension sites. The hybridization peak pairs under the Fermi level in Figs. 5b and 5d would reduce the total energies of the systems and stabilize the system structures. This also agrees well with the results of Bader analysis. This analysis has highlighted the importance of the orbital type of the valence electrons, and allows us to predict that the compression site geometry of the $\{10\bar{1}1\}$ CTB favors strong bonding with *p*-block elements

We have added the DOS analysis as a new paragraph in the revision. It reads:

“To verify the findings about chemical bonding in Bader analysis and to better understand the chemical bonding between the solute Bi, Gd or Zn and its nearby Mg atoms, electron density of states analysis was implemented to the cases where Bi (20% and 100% occupancy), Gd (100% occupancy), or Zn (100% occupancy) has segregated to the extension or compression site of the $\{10\bar{1}1\}$ CTB. Figure 5 plots partial density of states (PDOS) of the solute atom and its closest Mg atom. In Fig. 5b, two intense hybridization peak pairs between Bi-*p* and Mg-*s* are found at -5.6 eV and -2.4 eV when a Bi atom has segregated to the compression site of $\{10\bar{1}1\}$ with 20% occupancy, while no obvious bonding interaction is found for the case where the Bi has segregated to the extension site.

When the occupancy of Bi at the compression site reaches 100%, no hybridization occurs, as shown in Fig. 5c. For the case of Bi segregation to the extension site with 100% occupancy, the hybridization peaks move to the energy higher than Fermi energy, which might lead to a more unstable electronic structure. Similar to the case of Bi segregation (20% occupancy) at the compression site, the presence of the hybridization peak pairs in Figs. 5d indicates strong electronic interactions, when Gd segregated to the extension site, or Zn into the compression site (Fig. 5e). The hybridization peak pairs under Fermi level in Figs. 5b and 5d would reduce the total energies of the systems and stabilize the system structures. This also agrees well with the results of Bader analysis.”

2. The description of the computation of the differential electron charge density is a bit unclear. What is the starting point for the single iteration result that serves as the benchmark? Is this the overlap of atomic charges? What is the physics that makes this a good and reproducible choice.

The differential electron charge density (DECD) is computed by the following steps. a) Apply fully geometric relaxations to the studied supercell. b) Calculate the non-interaction charge density $\rho_{non-inter}(Mg_{N-m}X_m)$ by allowing only one electronic relaxation, which $\rho_{non-inter}(Mg_{N-m}X_m)$ as a reference reflects the states before the electron redistribution. c) Calculate the charge density $\rho_{inter}(Mg_{N-m}X_m)$ after fully electron relaxations. d) Calculate $\rho_{DECD}(Mg_{N-m}X_m)$ by subtracting the $\rho_{non-inter}(Mg_{N-m}X_m)$ from $\rho_{inter}(Mg_{N-m}X_m)$. The obtained $\rho_{DECD}(Mg_{N-m}X_m)$ represents the electron redistribution during the single atoms forming the structure of input supercell.

The single iteration result $\rho_{non-inter}(Mg_{N-m}X_m)$ represents the initial electron charge distribution without any electronic optimization. It can be thought as a simple overlap of atomic charges.

As stated above, DECD describes the electron charge redistribution during the optimized system formation. In DECD contour maps, the area with positive values represents the area with charge accumulation while the negative values indicate charge depletion. Herein, the ρ_{DECD} describes the valence electron transfer and can be used to analyze the bonding between involved atoms. At present, DECD is widely used in many simulation studies about diffusion or solute-defect interactions to investigate the electronic interaction between Mg atoms and solute atoms^{45, 46}.

In the revision, the description of DECD in the calculation methods has been modified. This part now reads:

“Differential electron charge density ρ_{DECD} of a fully relaxed supercell in this work is given by the electron charge with static self-converged electronic relaxation subtracting the charge density with only one step electronic relaxation:

$$\rho_{DECD}(Mg_{N-m}X_m) = \rho_{inter}(Mg_{N-m}X_m) - \rho_{non-inter}(Mg_{N-m}X_m) \quad (2)$$

where ρ_{inter} is the charge density of supercell after completely converged electronic relaxations, and $\rho_{non-inter}$ is the reference charge density (or non-interaction charge density) obtained by applying only one step electronic relaxation to the fully relaxed supercell. $N - m$ and m are the number of Mg atoms and solute atoms in the input supercell, respectively. The DECD is widely utilized in other simulation studies regarding diffusion or solute-defect interaction to investigate the interactions between Mg atoms and solute atoms^{45,46}.”

3. Also, the relaxation of the supercell is mentioned, but it is unclear to me which axes of the supercells were allowed to relax? If it was all axes, then I am a bit confused, as the lattice parameters in the plane of the tilt boundary should be fixed. (The tilt boundaries may necessarily change the lattice parameters, but this is an artifact of their extreme high density in the periodic supercell approach.)

The reviewer is correct to point out the high density of twin boundaries in the model. The original calculations were performed allowing all supercell axes to relax, which will relax some strain in the twin boundary at the expense of deforming a small amount of bulk Mg in the model. The calculations have now been re-performed in the revision, allowing only the lattice parameter perpendicular to the twin boundary to relax, with the other two lattice parameters set to values that constrain the Mg atoms to their bulk spacing in these directions. This more realistically simulates the twin boundary surrounded by a large number of Mg atoms. As shown in the revised Figure 3 and Figure S5, these new calculations have increased the strain in the twin boundary, and so have changed the energies of the reference solute-free twin boundary models, as well as the models with various solute concentrations. The general effect has been a slight destabilisation of the solute stabilities, now with Bi found to be unstable at all concentrations in the $\{10\bar{1}2\}$ in the extension site, consistent with our experimental observations.

4. Also, how was the lattice parameter for the alloys fixed? Did the authors use a bulk calculation of a random solid solution alloy to determine these parameters?

In the calculations of alloys systems, we still chose to apply full relaxations to solute-containing supercells which does not contain a CTB and allow all axes to relax.

We didn't use a bulk calculation of a random solid solution alloy to determine lattice parameter in the alloy system. Our simulation method is the same as most other first-principles calculations works regarding solute segregation on twin boundary. The initial state is considered to be that with one solute atom embedded in the Mg matrix and our twin boundary solute-free, while the final state is that the solute atom has diffused (segregated) into the site of the twin boundary. The original place of the solute atom in the matrix has been exchanged with the Mg atom at the site of the twin boundary, after which the twin boundary becomes solute-segregated and the Mg matrix contains no solute. The segregation energy is the energy change during the thermodynamic process. Thus, we didn't calculate the random solid solution alloy but relaxed the supercells needed by the logic above. In this manner, the segregation energy value of Gd or Zn segregation into $(10\bar{1}2)$ coherent twin boundary was calculated to be -0.34 eV or -0.27 eV, which were very close to those from the previous works^{13,23}.

Reviewer #3

1. There is one small question regarding the appearance of the 10-11 twin boundary in Mg-Pb (Suppl. Fig. 7): it appears that a faceting of the twin boundary is present. Do the authors think this could be resulting from the Pb segregation to minimize the elastic strain energy induced by Pb segregation or do they interpret this differently?

The faceting of the $\{10\bar{1}1\}$ twin boundary is actually a twin step formed on the otherwise coherent twin boundary. It has the character of dislocation and is usually called ‘twinning dislocation’, according to the definition of Pond *et al.*^{R8}, whose gliding on the twin plane results in either growth or shrinkage of the twin. We think that the step shown in Fig. R2a was not induced by Pb segregation but was related with the twinning process. This is because such a step, $S_{3/2}$ where the subscripts 3 and 2 represent the number of $\{10\bar{1}1\}$ planes associated with the step in the matrix and the twin region respectively, could also be found in the deformed sample without any annealing or ageing treatments (Fig. R2b). Furthermore, the occurrence of $S_{3/2}$ step was also reported in other deformed HCP metals such as pure Ti^{R9}.

Figure R2. HAADF-STEM images showing $S_{3/2}$ steps on $\{10\bar{1}1\}$ twin boundaries in samples of Mg–1.5at.%Pb alloy. (a) Pb segregation is revealed in a $\{10\bar{1}1\}$ coherent twin boundary of deformed and aged sample (17% compression + 200 °C/5 hours). (b) Absence of Pb segregation in a $\{10\bar{1}1\}$ coherent twin boundary of the deformed sample without any artificial ageing treatment (17% compression).

References

- R1. Seah, M. P. & Hondros, E. D., A. Grain boundary segregation. *Proc. R. Soc. Lond. A* **335**, 191-212, (1973).
- R2. Reed-Hill, R. E. & Robertson, W. D. Additional modes of deformation twinning in magnesium. *Acta Metall.* **5**, 717-727, (1957).
- R3. Hartt, W. H. & Reedhill, R. E. The irrational habit of second-order $(10\bar{1}1)$ - $(10\bar{1}2)$ twins in magnesium. *Trans. Met. Soc. AIME* **239**, 1511-&, (1967).
- R4. Yu, Q., Jiang, Y. & Wang, J. Cyclic deformation and fatigue damage in single-crystal magnesium under fully reversed strain-controlled tension–compression in the $[10\bar{1}0]$ direction. *Scr. Mater.* **96**, 41-44, (2015).
- R5. Ando, D., Koike, J. & Sutou, Y. Relationship between deformation twinning and surface step formation in AZ31 magnesium alloys. *Acta Mater.* **58**, 4316-4324, (2010).
- R6. Reedhill, R. E. A study of the $(10\bar{1}1)$ and $(10\bar{1}3)$ twinning modes in magnesium. *Trans. Am. Inst. Min. Metall. Eng.* **218**, 554-558, (1960).
- R7. Wang, Y., Chen, L. Q., Liu, Z. K. & Mathaudhu, S. N. First-principles calculations of twin-boundary and stacking-fault energies in magnesium. *Scr. Mater.* **62**, 646-649, (2010).
- R8. Pond, R. C., Hirth, J. P., Serra, A. & Bacon, D. J. Atomic displacements accompanying deformation twinning: shears and shuffles. *Mater. Res. Lett.* **4**, 185-190, (2016).
- R9. Li, Y. J. *et al.* Faceted interfacial structure of $\{10\bar{1}1\}$ twins in Ti formed during equal channel angular pressing. *Scr. Mater.* **62**, 443-446, (2010).

REVIEWERS' COMMENTS

Reviewer #1 (Remarks to the Author):

The authors have responded admirably to the concerns of the reviewers. They have taken seriously each comment, performed additional analysis, and modified the manuscript accordingly. I have no further comment, other than to say, I wish the readers could see the helpful commentary in the "response to reviewers" document, in addition to the manuscript itself.

I have no further reservations.

Reviewer #2 (Remarks to the Author):

I believe that the author's have answered my initial criticisms (and those of the other reviewers) quite well. I do think that the results they find are of interest, and do point to the specific chemical interactions as being dominant.

I believe that the manuscript will be of interest to the readers of Nature Communications, and that it should be published.